# The prevalence of hyperglycemia and its impact on mortality among people living with HIV in Georgia

Tea Borkowska[1]*, Nikoloz Chkhartishvili[1], Ekaterine Karkashadze[1], Otar Chokoshvili[1], Pati Gabunia[1], Lali Sharvadze[1,2], Tengiz Tsertsvadze[1,2]

1 Infectious Diseases, AIDS & Clinical Immunology Research Center, Tbilisi, Georgia, 2 Ivane Javakhishvili Tbilisi State University, Tbilisi, Georgia

* Theanozadze@yahoo.com

## Abstract

**Data Availability Statement:** All relevant data are within the paper and its Supporting information files.

### Background

Life expectancy and quality of life of people living with HIV have been dramatically improved after introducing antiretroviral therapy, and the prevalence of non-communicable diseases has increased. Several studies have found that hyperglycemia with or without type 2 diabetes was associated with poor outcomes in people living with HIV.

The study's objective was to determine the prevalence of hyperglycemia and assess its impact on mortality.

### Materials and methods

A retrospective cohort study was conducted among people living with HIV diagnosed in 2012–2018 and followed through 2020 at the Infectious Diseases, AIDS and Clinical Immunology Research Center in Tbilisi, Georgia. Primary outcomes of interest included the prevalence of hyperglycemia and mortality. Causes of death were classified according to the Coding of Death in HIV (CoDe) protocol.

### Results

Our study included 2914 people living with HIV. Two hundred and forty-two (8.3%) patients had hyperglycemia, with an increasing prevalence by age. Three hundred one (9.7%) participants died over the median 3.71 (IQR: 2.14–5.37) years of follow-up. Among these, 139 (46.2%) were due to AIDS- related causes, 123 (40.9%)—were due to non-AIDS causes, and in 39 (12.9%) cases, the cause of death could not be determined. Overall, the cohort contributed to 11,148 person-years of follow-up (PYFU), translating into a mortality rate of 2.70 deaths per 100 PYFU. The mortality rate was significantly higher among individuals with hyperglycemia—11.17 deaths per 100 PYFU vs 2.07 deaths per 100 PYFU among normoglycemic patients(p<0.0001).

**Funding:** The author(s) received no specific funding for this work.

**Competing interests:** The authors have declared that no competing interests exist.

## Conclusions

Hyperglycemia was associated with increased odds of mortality. Screening and management of hyperglycemia should be integrated into routine HIV clinical services as part of a comprehensive care package.

## Introduction

Life expectancy and quality of life of people living with HIV (PLWH) have been dramatically improved after introducing antiretroviral therapy (ART) [1]. The longer people living with HIV are on ART, the more likely they are to develop non-communicable diseases, especially metabolic disorders [2–6].

It has been shown that HIV infection and ART increase the risk of metabolic disorders, abnormal glucose metabolism, and type 2 diabetes (T2D). The development of hyperglycemia and T2D during HIV depends on many factors, such as HIV infection's duration, degree of immunosuppression, and exposure to ARV medicaments [7]. Specific ARV medicaments, mainly protease Inhibitors (PI), nucleoside Reverse Transcriptase Inhibitors (NRTI), and some non-Nucleoside Reverse Transcriptase Inhibitors (NNRTI-s), like efavirenz, are associated with T2D [8,9], dysglycemia, dyslipidemia, lipodystrophy, arterial hypertension, and myocardial infarction [7,8]. Longer ART exposure is linked to a higher prevalence of hyperglycemia and diabetes [2–4]. The prevalence of T2D in people living with HIV is two-fold higher [4,7] and ranges from 2% to 14% [2]. With or without T2D, hyperglycemia increases morbidity and mortality among people with infectious diseases, including tuberculosis and HIV [5,7,10–14]. Diabetes increases mortality risk due to circulatory system and kidney complications, which are significant causes of excess mortality in people living with HIV. After considering this dramatic effect, current guidelines recommend routine screening for glucose with appropriate management, including early detection and treatment of pre-diabetes and diabetes [15]. However, few programs offer integrated HIV/non-communicable disease management in resource-limited countries. There is a gap in data on the burden of non-communicable diseases among people living with HIV [16].

Georgia lacks knowledge regarding the relationship between hyperglycemia and mortality in people living with HIV. The study's objective was to determine the prevalence of hyperglycemia and assess its impact on mortality.

## Materials and methods

A retrospective cohort study was conducted among people living with HIV receiving care at the Infectious Diseases, AIDS and Clinical Immunology Research Center in Tbilisi, Georgia, the country's referral institution for HIV providing care to 64% of people living with HIV. The study included adults (age ≥18 years) living with HIV diagnosed in 2012–2018 and followed through 2020. Patients who died or were lost to follow-up within six months of diagnosis (3.7%) were excluded from the analysis.

The study used routinely available data collected as part of the standard of care. All data, including demographic, clinical, and laboratory information, was extracted from the national AIDS health information system (AIDS HIS), a secure web-based database collecting data on all confirmed HIV cases.

During the study period, national HIV guidelines recommended checking for glycemia at the entry into HIV care and then annually. Given that care for endocrinologic disorders was

not integrated into HIV care, patients with hyperglycemia were referred to relevant specialists for further management. Therefore, AIDS HIS contained information only on glycemia. Glucose levels used for the statistical analysis were checked before ART initiation.

American Diabetes Association criteria defined hyperglycemia as a fasting glucose level> 5.6 mmol/l. or postprandial glucose level >7.8 mmol/l [17]. According to International Diabetes Federation (IDF), postprandial and postmeal plasma glucose levels should not rise above 7.8 mmol/l in healthy people [18]. Glucose measurements were considered postprandial if it was unknown whether it was measured during fasting or not.

## Statistical analysis

Primary outcomes of interest included the prevalence of hyperglycemia and mortality (including all-cause mortality and cause-specific mortality). Causes of death were classified according to the Coding of Death in HIV (CoDe) protocol [19]. Based on CoDe protocol, deaths were classified into the following categories: AIDS-related deaths, non-AIDS and unknown.

The descriptive analysis was performed using the median and interquartile range for continuous variables and frequency/percentage for categorical data. Bivariate comparisons were tested using Pearson's chi-square test. Time-to-event approach was used for deriving mortality rates calculated as the number of deaths divided by the number of total person-years of follow-up (PYFU) contributed to the observation period. Kaplan-Meier method was used to estimate the probability of survival by hyperglycemic status. The association between hyperglycemia and mortality was assessed in Cox proportional hazards regression analysis with a competing risk model used for estimating the cause-specific adjusted hazard ratios [20]. In the sensitivity analysis, we evaluated the influence of hyperglycemia on all-cause and cause-specific mortality by CD4 cell strata (people with CD4 count <200 and ≥200 cells/mm$^3$). A p-value <0.05 was considered statistically significant. All analyses were conducted using SAS 9.4 (SAS Institute, Cary, NC, USA).

## Ethics statement

The study was approved by the Institutional Review Board of the Infectious Diseases, AIDS and Clinical Immunology Research Center (OHRP #: IRB00006106). All patients initiating HIV care provide informed consent on the use of information collected as part of routine clinical care for research. The study used secondary data extracted from the national HIV database as an anonymous dataset with no personal identities. Therefore, informed consent was waived by IRB.

## Results

Our study included 2914 people living with HIV. The median age was 36 (IQR: 28–45) years, and 2205 (75.7%) were men. Overall, 619 (21.2%) participants had a CD4 count of ≤200 cells/mm$^3$ at HIV diagnosis. AIDS was documented in 1189 (40.8%) patients. Every single patient has begun ART. Comorbidities, including cardiovascular diseases, kidney disease, and cancer, were present in 183 (6.3%) cases (Table 1).

The information about fasting glucose was unknown in 18.6% of cases. Two hundred and forty-two (8.3%) patients had hyperglycemia, increasing prevalence by age. People with baseline CD4 count <200 cells/mm$^3$ and a history of AIDS diagnosis had a statistically significantly higher prevalence of hyperglycemia (Table 1). The distribution of hyperglycemia in people living with HIV did not differ significantly by gender.

A total of 301 (9.7%) people living with HIV died over the median 3.71 (IQR: 2.14–5.37) years of follow-up. Among these 301 death events, 139 (46.2%) were due to AIDS-related

**Table 1. Study population characteristics and prevalence of hyperglycemia by baseline characteristics (n = 2914).**

| Characteristic | Total N | Hyperglycemia, n (%) | p-value |
|---|---|---|---|
| **Total cohort** | 2914 | 242 (8.3) | |
| **Age at HIV diagnosis** | | | |
| 18<30 | 868 | 28 (3.2) | <0.0001 |
| 30-<40 | 947 | 65 (6.9) | |
| 40-<50 | 716 | 82 (11.5) | |
| 50-<65 | 357 | 62 (17.4) | |
| 65+ | 26 | 5 (19.2) | |
| **Gender** | | | |
| Men | 2205 | 193 (8.7) | 0.12 |
| Women | 709 | 49 (6.9) | |
| **Baseline CD4 cell count** | | | |
| <200 | 619 | 98 (15.8) | <0.0001 |
| 200–500 | 1085 | 82 (7.6) | |
| >500 | 1210 | 62 (5.1) | |
| **AIDS diagnosis** | | | |
| Yes | 1189 | 169 (14.2) | <0.0001 |
| No | 1725 | 73 (4.2) | |
| **Non-communicable comorbidities** | | | |
| With comorbidities | 183 | 50 (27.3) | <0.0001 |
| Without comorbidities | 2712 | 192 (7.0) | |

causes, 123 (40.9%)—were due to non-AIDS causes, and in 39 (12.9%) cases, the cause of death could not be determined.

Overall, the cohort contributed to 11,153 person-years of follow-up (PYFU), translating into a mortality rate of 2.70 deaths per 100 PYFU. The mortality rate was significantly higher among individuals with hyperglycemia—11.17 deaths per 100 PYFU vs. 2.07 deaths per 100 PYFU among normoglycemic patients (p<0.0001). Kaplan-Meier survival estimates showed statistically significant differences between hyperglycemic and normoglycemic groups for all-cause and cause-specific mortality (Fig 1).

Hyperglycemia remained significantly associated with increased hazards of death for all-cause mortality (HR: 2.32, 95% CI: 1.76–3.04. p<0.0001), for AIDS-related mortality (HR: 2.24, 95% CI: 1.50–3.35, p<0.0001) and non-AIDS mortality (HR: 1.93, 95% CI: 1.18–3.14, p = 0.008) in multivariate regression analysis (Table 2). In sensitivity analysis, hyperglycemia was associated with increased hazards of all-cause mortality in both people with CD4 count <200 cells/mm$^3$ (HR: 2.02, 95% CI: 1.47–2.78, p<0.0001) and CD4 count ≥200 cells/mm$^3$ (HR: 3.26, 95% CI: 1.96–5.42, p<0.0001). Regarding cause-specific mortality, among people with CD4 count <200 cells/mm$^3$, hyperglycemia was associated with AIDS-related mortality but not with non-AIDS deaths, and vice versa among persons with CD4 count >200 cells/mm$^3$ hyperglycemia was associated with non-AIDS mortality and not with AIDS-related mortality (Table 3). Hyperglycemia did not show a significant association with mortality from unknown causes in any multivariate regression model.

## Discussion

Hyperglycemia is common among people living with HIV, which has a detrimental effect on survival. In our study, hyperglycemia was associated with a more than 2-fold increase in mortality risk. Association between diabetes mellitus and death has been well documented [21],

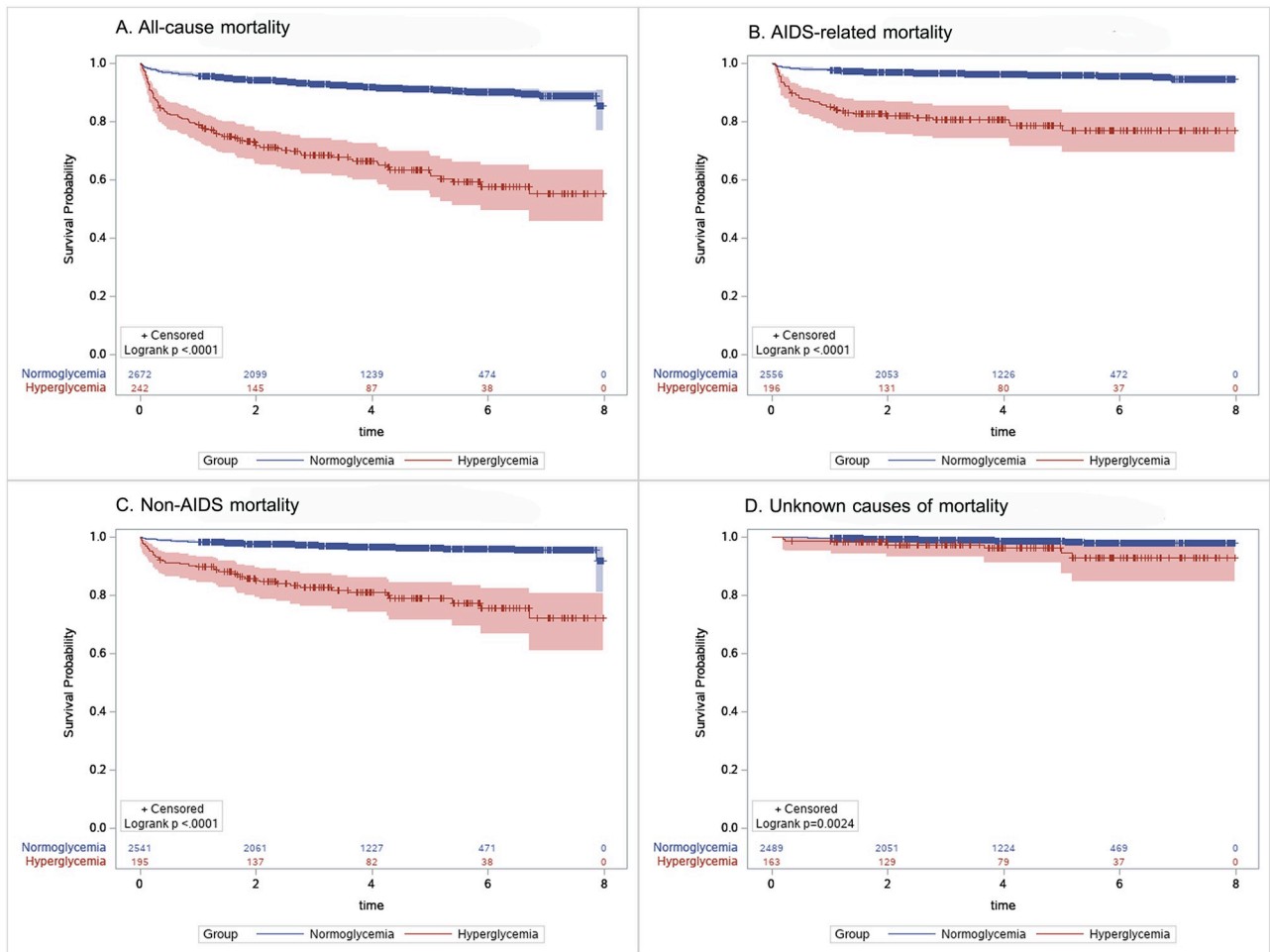

**Fig 1. Kaplan-Meier survival estimates for all-cause and cause-specific mortality with number of participants at risk and 95% confidence intervals.**

and our findings add to the body of knowledge showing that hyperglycemia is an independent risk factor for mortality [11,13,22,23]. Hyperglycemia can indicate missed diagnosis of diabetes mellitus, especially in resource-limited settings where care for non-communicable diseases is rarely integrated into HIV service delivery [24]. HIV treatment programs in resource-limited countries, including Eastern Europe, are primarily focused on acute care of HIV, mainly because of high rates of late diagnosis resulting in excess morbidity and mortality [25]. Comprehensive screening and treatment for diabetes and pre-diabetes are required to improve disease outcomes [26].

The overall prevalence of hyperglycemia was 8.3%, which is lower than studies conducted in China [2] and Africa [27], where the prevalence of hyperglycemia was 19.99% and 12.4%, respectively. The prevalence of hyperglycemia was significantly higher in older people, which is not surprising because hyperglycemia and diabetes mellitus are more common in the older population, and the prevalence increases with age [17]. Many other studies have found a similar relationship between increasing age and the prevalence of T2D among people living with HIV [5,28,29]. It should be taken into account that our cohort was relatively young, with a median age of 36 years, which might have influenced the overall prevalence. Furthermore, weight gain, obesity and insulin resistance can increase blood glucose levels [30]. The high

**Table 2. Multivariate analysis of the association of hyperglycemia with all-cause and cause-specific mortality (n = 2914).**

| | Unadjusted | | Adjusted | |
|---|---|---|---|---|
| | Hazard ratio (95% CI) | p-value | Hazard ratio (95% CI) | p-value |
| **All-cause mortality** | | | | |
| Hyperglycemia | 5.14 (4.00–6.60) | <0.0001 | 2.32 (1.76–3.04) | <0.0001 |
| Normoglycemia | 1 | | 1 | |
| **AIDS-related mortality** | | | | |
| Hyperglycemia | 4.71 (3.26–6.81) | <0.0001 | 2.24 (1.50–3.35) | <0.0001 |
| Normoglycemia | 1 | | 1 | |
| **Non-AIDS mortality** | | | | |
| Hyperglycemia | 5.40 (3.69–7.88) | <0.0001 | 1.93 (1.18–3.14) | 0.008 |
| Normoglycemia | 1 | | 1 | |
| **Mortality from unknown cause** | | | | |
| Hyperglycemia | 2.34 (1.04–5.28) | 0.04 | 1.72 (0.69–4.26) | 0.24 |
| Normoglycemia | 1 | | 1 | |

The multivariate regression model was adjusted for age, gender, HIV transmission mode, baseline CD4 cell count, history of AIDS diagnosis, and non-communicable comorbidities.

prevalence of hyperglycemia among people with comorbidities is not surprising, given the well-known association between diabetes and the risk of other non-communicable diseases [30,31].

Our study was not powered to explain the association found between hyperglycemia and advanced HIV disease (low CD4 cell count, AIDS diagnosis). Similar associations have been reported from China and Thailand, suggesting that hyperglycemia might be connected with ART and HIV itself, especially with uncontrolled HIV [2,32].

HIV can cause chronic inflammation [33,34]. As a result, the concentration of pro-inflammatory cytokines, such as tumour necrosis factor (TNF)-α, IL-6, IL-8 and IL-18 increases. Insulin resistance, metabolic syndrome and T2D are associated with elevated IL-18 levels [35]. During HIV, systemic inflammation and tryptophan catabolism alert the gut microbiota. As a

**Table 3. Multivariate analysis of the association of hyperglycemia with all-cause and cause-specific mortality by baseline CD4 cell count (n = 2914).**

| | CD4 <200 (n = 558) | | CD4 ≥200 (n = 2,356) | |
|---|---|---|---|---|
| | Adjusted Hazard ratio (95% CI) | p-value | Adjusted Hazard ratio (95% CI) | p-value |
| **All-cause mortality** | | | | |
| Hyperglycemia | 2.02 (1.47–2.78) | <0.0001 | 3.26 (1.96–5.42) | <0.0001 |
| Normoglycemia | 1 | | 1 | |
| **AIDS-related mortality** | | | | |
| Hyperglycemia | 2.47 (1.63–3.75) | <0.0001 | 0.84 (0.19–3.67) | 0.82 |
| Normoglycemia | 1 | | 1 | |
| **Non-AIDS mortality** | | | | |
| Hyperglycemia | 1.16 (0.65–2.06) | 0.62 | 4.60 (2.28–9.26) | <0.0001 |
| Normoglycemia | 1 | | 1 | |
| **Mortality from unknown cause** | | | | |
| Hyperglycemia | 1.30 (0.40–4.27) | 0.66 | 2.36 (0.59–9.37) | 0.22 |
| Normoglycemia | 1 | | 1 | |

The multivariate regression model was adjusted for age, gender, HIV transmission mode, history of AIDS diagnosis, and non-communicable comorbidities.

result, *Bifidobacterium*, *Bacteroides*, *Clostridium*, *Anaerovibri*, *Akkermansia*, *Finegoldia*, *Anaerococcus*, *Faecalibacterium*, and *Roseburia* concentrations are significantly lower among people living with HIV. *Anaerococcus* produce butyrate, which has an anti-inflammatory effect. Moreover, butyrate improves insulin sensitivity and reduces the risk of diabetes. Based on the aforesaid, low *Anaerococcus* concentration alerts metabolism and leads to metabolic syndrome and T2D [33,34,36,37]. The higher prevalence of hyperglycemia in people with advanced HIV should be interpreted in light of previous studies that showed an association between a low CD4 cell count and non-AIDS events [38,39].

Our findings align with previous reports indicating increased mortality among persons with hyperglycemia. While some of the reports suggest an increased risk of death among people living with HIV with serious opportunistic infections (e.g., tuberculosis, cryptococcal meningitis) [11,13,22], our data suggest the negative impact of hyperglycemia on survival irrespective of HIV disease status. After adjusting for CD4 cell count and history of AIDS diagnosis, hyperglycemia was significantly associated with all-cause and cause-specific mortality. Moreover, sensitivity analyses, stratified by CD4 cell count, further confirmed the negative effect of hyperglycemia on all cause-mortality. Increased risk of AIDS-specific mortality among people with low baseline CD4 cell count could suggest a reciprocal relationship between hyperglycemia and advanced HIV disease, which should be considered in light of high rates of late HIV diagnosis and serious AIDS-defining conditions in the country [40,41].

Our study has limitations. The major limitation is the potential underestimation of hyperglycemia because all glucose measurements with unknown fasting information were considered postprandial. We did not have information on diabetes diagnosis. Some patients might have been on anti-diabetic drugs and thus presented with normal glucose levels. Another potential source of misclassification is the use of single-point random glucose levels may cause misclassification of some patients as hyperglycaemic without further confirmation using OGTT or HbA1c. As a result of this study, these tests are being implemented as part of routine care. Also, this study did not assess other factors, such as obesity/high BMI and lifestyle, that may contribute to hyperglycemia/impaired fasting glucose.

Despite these limitations, the study provides essential information to address the growing problem of non-communicable diseases among people living with HIV. This was the first study to assess the prevalence and impact of hyperglycemia in a nationally representative sample of people living with HIV in Georgia and the entire Eastern European region. Our research showed that hyperglycemia is common among people living with HIV, resulting in increased mortality risk. Screening and management of hyperglycemia should be integrated into routine HIV clinical services as part of a comprehensive care package.

## Supporting information

**S1 File. Dataset.**
(XLSX)

## Acknowledgments

The authors are grateful to all people involved in the implementation of national HIV/AIDS programs.

## Author Contributions

**Conceptualization:** Tea Borkowska, Nikoloz Chkhartishvili.

**Data curation:** Tea Borkowska, Nikoloz Chkhartishvili, Ekaterine Karkashadze, Otar Chokoshvili.

**Formal analysis:** Nikoloz Chkhartishvili, Ekaterine Karkashadze, Otar Chokoshvili.

**Investigation:** Tea Borkowska, Nikoloz Chkhartishvili.

**Methodology:** Tea Borkowska.

**Project administration:** Tea Borkowska, Pati Gabunia, Lali Sharvadze, Tengiz Tsertsvadze.

**Resources:** Tea Borkowska.

**Software:** Nikoloz Chkhartishvili, Ekaterine Karkashadze, Otar Chokoshvili.

**Supervision:** Tea Borkowska, Nikoloz Chkhartishvili, Pati Gabunia, Lali Sharvadze, Tengiz Tsertsvadze.

**Visualization:** Tea Borkowska, Ekaterine Karkashadze.

**Writing – original draft:** Tea Borkowska, Nikoloz Chkhartishvili.

**Writing – review & editing:** Tea Borkowska, Nikoloz Chkhartishvili, Ekaterine Karkashadze, Otar Chokoshvili, Pati Gabunia, Lali Sharvadze, Tengiz Tsertsvadze.

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
