## [Decision Letter · Decision Letter 0]

27 Sep 2022

PONE-D-22-13596

The prevalence of hyperglycemia and its impact on mortality among people living with HIV in Georgia

PLOS ONE

Dear Dr. Borowska,

Thank you for submitting your manuscript to PLOS ONE. After careful consideration, we feel that it has merit but does not fully meet PLOS ONE’s publication criteria as it currently stands. Therefore, we invite you to submit a revised version of the manuscript that addresses the points raised during the review process.

We look forward to receiving your revised manuscript.

Kind regards,

Alana T Brennan

Academic Editor

PLOS ONE

“Many thanks to all the doctors and staff of the Infectious Diseases, AIDS & Clinical Immunology Research Center who support the implementation of the HIV/AIDS Program”

“The study has been conducted with internal resources of Infectious Diseases, AIDS & Clinical Immunology Research Center”

“The authors have no conflicts of interest to disclose.”

Reviewers' comments:

Reviewer's Responses to Questions

**Comments to the Author**

1. Is the manuscript technically sound, and do the data support the conclusions?

Reviewer #1: Yes

Reviewer #2: Yes

2. Has the statistical analysis been performed appropriately and rigorously? 

Reviewer #1: Yes

Reviewer #2: Yes

3. Have the authors made all data underlying the findings in their manuscript fully available?

Reviewer #1: No

Reviewer #2: Yes

4. Is the manuscript presented in an intelligible fashion and written in standard English?

Reviewer #1: Yes

Reviewer #2: Yes

5. Review Comments to the Author

Reviewer #1: Increasing prevalence of noncommunicable diseases among people living with HIV is of growing concern globally. The authors aim to characterize the presence of hyperglycemia and its associated risk of mortality among a large cohort of HIV positive individuals in Georgia. Overall, these results fill an important knowledge gap in the current literature and should be published. However, authors should address or take into consideration the comments below:

1) In the introduction, the authors clearly establish an association between specific classes of ART regimens - including PIs, NRTIs, and some NNRTIs - and ART duration with type 2 diabetes. Do the authors have any data on regimen type and/or duration? If so, given this established increased risk of hyperglycemia among people living with HIV with longer exposure to ART and on certain regimens, authors should present median ART duration and ART regimen class in Table 1. If it is found that ART duration and/or class has a statistically significant relationship with hyperglycemia, authors should adjust for these variables in the multivariate analyses.

2) Authors state a major limitation of the paper is the potential underestimation of hyperglycemia because all glucose measurements with unknown fasting information were considered postprandial. In order to give the reader a sense of how much this may have impacted your results, authors should report what % of glucose measurements had unknown fasting information.

3) Per journal requirement, ethics statement needs to be included in the methods section of the manuscript.

4) Per journal requirement, authors need to make all raw data available or provide an explanation to data restrictions.

Reviewer #2: Overall: This is an interesting paper about the prevalence of hyperglycaemia and its impact on people living with HIV. It well-written paper, however, there are some problems with definition of hyperglycemia that need addressing. It highlights the importance of screening for dysglycaemia in individual living with HIV at the entry into care and as a part continually assessment.

Revision

1. Major changes

Line 90-92: Authors should justify why impaired fasting glucose levels were used to define hyperglycaemia. According to the guidelines, the postprandial glucose level > 7.8 mmol/L was based on an oral glucose tolerance test (OGTT) which is not highlighted in this paper.

The use of single point random glucose level may cause a misclassification of some patients as hyperglycaemic. A single high glucose level is less definitive (compared to OGTT or HbA1c) to classify dysglycemia. Consider highlighting this as a limitation of the study.

The authors should clarify if the glucose level considered for analysis was at the entry into HIV care as recommended by National guidelines or at any point during follow-up (or before ART initiation). Because both HIV and ART are implicated in development of hyperglycaemia/diabetes.

2. Minor changes

Abstract

Line 37: Please change 242 to Two hundred and forty-two

Line 38: Also, change 301 to Three hundred one

Introduction

Line 56: Change Type to type 2 diabetes

Line 59: Please change Efavirenz to efavirenz.

Line 59: Write PI, NRTI, and NNRT in full and abbreviate thereafter.

Method

Line 81: What is the percentage of patients that were excluded from the analysis due to loss of follow-up?

Line 93: rewrite ifit as if it

Authors should add the under method that the approval was obtained from the Ethics Committee of the Infectious Disease, AIDS, & Clinical Immunology Research Center.

Results

Line 120-121 (table 1):

• Please update the percentages for the population (total cohort) with hyperglycaemia to 8.3% and adjust accordingly throughout the manuscript including the abstract.

• Age category (30-39)- change 5.3 to 5.4%

• Age category (65+) – change to 25.8 %

Discussion

Line 178-179: The authors should explain the possible reason why they found a lower prevalence of hyperglycaemia compared to other published studies.

Line 190: Consider rephrasing ‘our study cannot be clearly explained’ statement.

Line 192-193: ‘The higher prevalence of hyperlycemia in people with advanced HIV should be interpreted considering previous findings indicating an association between low CD4 count and non-AIDS event’. It sounds there missing phrase? Consider rephrasing it.

Under discussion, authors should add the mechanism linking HIV to hyperglycaemia

Other factors such as obesity/ high BMI, and lifestyle that may contribute to hyperglycaemia/impaired fasting glucose were not assessed in this study. Should be highlighted as a limitation.

General comment:

Authors should leave the space between the last word and reference consistent throughout the manuscript.

Modify the font type for references

6. PLOS authors have the option to publish the peer review history of their article (what does this mean?). If published, this will include your full peer review and any attached files.

Reviewer #1: **Yes: **Emma M. Kileel

Reviewer #2: No

---

## [Author Response · Author response to Decision Letter 0]

5 Oct 2022

Dear Dr. Alana T Brennan

We appreciate the opportunity to revise our manuscript for further review at Plos One. The manuscript has been substantially revised in accordance with comments provided by Editorial Board and reviewers. The manuscript has been formatted by the instruction provided. Below are point-by-point responses to the Academic Editor and reviewers’ comments:

Reviewer #1 (Emma M. Kileel): Increasing prevalence of noncommunicable diseases among people living with HIV is of growing concern globally. The authors aim to characterize the presence of hyperglycemia and its associated risk of mortality among a large cohort of HIV positive individuals in Georgia. Overall, these results fill an important knowledge gap in the current literature and should be published. However, authors should address or take into consideration the comments below:

Comment: In the introduction, the authors clearly establish an association between specific classes of ART regimens - including PIs, NRTIs, and some NNRTIs - and ART duration with type 2 diabetes. Do the authors have any data on regimen type and/or duration? If so, given this established increased risk of hyperglycemia among people living with HIV with longer exposure to ART and on certain regimens, authors should present median ART duration and ART regimen class in Table 1. If it is found that ART duration and/or class has a statistically significant relationship with hyperglycemia, authors should adjust for these variables in the multivariate analyses.

Response: Dear Dr. Emma M Kileel, I want to express my gratitude for your time, suggestions, and advice. Glucose levels considered for analysis were at the entry into HIV care, as recommended by National guidelines (Before ART initiation). Every patient has begun the ART regimen at some point (some faster, some later). The median ART duration was 3.58 (2.86). We did not adjust variables in the multivariate analysis because all our glucose measurements were taken before the ART regimen. Information about ART and mortality is not in the tables either because every patient is on ART, and we would not be able to find a statistically significant difference. 

Comment: Authors state a major limitation of the paper is the potential underestimation of hyperglycemia because all glucose measurements with unknown fasting information were considered postprandial. In order to give the reader a sense of how much this may have impacted your results, authors should report what % of glucose measurements had unknown fasting information.

Response: Based on the given data, information about fasting glucose was unknown in 18.6% of cases. This information will be included in the paper as it was suggested. Thank you for your comment and advice.

Comment: Per journal requirement, an ethics statement needs to be included in the methods section of the manuscript.

Response: I added the following information in the method section: The study was approved by the Institutional Review Board of the Infectious Diseases, AIDS and Clinical Immunology Research Center (OHRP #: IRB00006106). All patients initiating HIV care provide informed consent on the use of information collected as part of routine clinical care for research. The study used secondary data extracted from the national HIV database as an anonymous dataset with no personal identities. Therefore, informed consent was waived by IRB.

Comment: Per journal requirement, authors need to make all raw data available or provide an explanation to data restrictions.

Response: Thank you very much for pointing it out. All raw data is uploaded on protocols.io 

URL: https://www.protocols.io/private/F851E67044A111EDB19C0A58A9FEAC02

DOI: dx.doi.org/10.17504/protocols.io.3byl4jke8lo5/v1

Thank you very much for your help.

Reviewer #2: Overall: This is an interesting paper about the prevalence of hyperglycaemia and its impact on people living with HIV. It well-written paper, however, there are some problems with definition of hyperglycemia that need addressing. It highlights the importance of screening for dysglycaemia in individual living with HIV at the entry into care and as a part continually assessment.

Revision

1. Major changes

Comment: Line 90-92: Authors should justify why impaired fasting glucose levels were used to define hyperglycaemia. According to the guidelines, the postprandial glucose level > 7.8 mmol/L was based on an oral glucose tolerance test (OGTT) which is not highlighted in this paper.

Response: According to IDF (International Diabetes Federation), postprandial and postmeal plasma glucose levels should not rise above 7.8 mmol/l in healthy people. I am attaching the link below https://www.in.gov/health/files/Guideline_PMG_final.pdf ( page 7). Anything above 7.8 mmol/l is not considered as a normal range. I mentioned postmeal plasma glucose level and IDF in the text, as it was suggested. Thank you very much for your comment and help in clarifying the article.

Comment: The use of single point random glucose level may cause a misclassification of some patients as hyperglycaemic. A single high glucose level is less definitive (compared to OGTT or HbA1c) to classify dysglycemia. Consider highlighting this as a limitation of the study.

Response: The following statement, as added in recommendations “The use of single point random glucose level may cause misclassification of some patients as hyperglycaemic. To classify dysglycemia, a single high glucose level is less definitive (compared to OGTT or HbA1c). Unfortunately, OGGT or HbA1c has not been checked at the entry into HIV care; that’s why we did not have information about these variables. “ Thank you very much for your thoughtful comments.

Comment: The authors should clarify if the glucose level considered for analysis were at the entry into HIV care as recommended by National guidelines or at any point during follow-up (or before ART initiation). Because both HIV and ART are implicated in development of hyperglycaemia/diabetes.

Response: Glucose levels considered for analysis were at the entry into HIV care, as recommended by National guidelines (Before ART initiation). Information was added in the article as suggested. Thank you for your advice and suggestion.

2. Minor changes

Abstract:

Comment: 1. Line 37: Please change 242 to Two hundred and forty-two

Response: Line 37: 242 was changed to Two hundred and forty-two. Thanks for the suggestion.

Comment:2. Line 38: Also, change 301 to Three hundred one

Response: Line 38: 301 was changed to Three hundred one. Thanks for the suggestion.

Introduction

Comment: Line 56: Change Type to type 2 diabetes

Response: Line 56: Type was changed to type 2 diabetes 

Comment: Line 59: Please change Efavirenz to efavirenz.

Response: Line 59: Efavirenz was changed to efavirenz.

Comment: Line 59: Write PI, NRTI, and NNRT in full and abbreviate thereafter.

Response: Line 59: PI, NRTI, and NNRT were changed into protease Inhibitors (PI), nucleoside Reverse Transcriptase Inhibitors (NRTI), and some non-Nucleoside Reverse Transcriptase Inhibitors (NNRTI-s). Thank you very much for your suggestions.

Method

Comment: Line 81: What is the percentage of patients that were excluded from the analysis due to loss of follow-up?

Response: Line 81: 112 people (3.7%) were excluded from the analysis. I added it in methods, where I mentioned loss to follow-up. Thank you very much.

Comment: Line 93: rewrite ifit as if it

Response: Line 93: ifit was changed into if it.

Comment: Authors should add the under a method that the approval was obtained from the Ethics Committee of the Infectious Disease, AIDS, & Clinical Immunology Research Center.

Response: For more clarification, I added the following information: The study was approved by the Institutional Review Board of the Infectious Diseases, AIDS and Clinical Immunology Research Center (OHRP #: IRB00006106). All patients initiating HIV care provide informed consent on the use of information collected as part of routine clinical care for research. The study used secondary data extracted from the national HIV database as an anonymous dataset with no personal identities. Therefore, informed consent was waived by IRB.

Results

Comment: Line 120-121 (table 1):

• Please update the percentages for the population (total cohort) with hyperglycaemia to 8.3% and adjust accordingly throughout the manuscript including the abstract.

• Age category (30-39)- change 5.3 to 5.4%

• Age category (65+) – change to 25.8 %

Response: Table 1 has been updated. We decided to recheck our data once again to modify it. Thank you very much for your time and advice.

Discussion

Comment: Line 178-179: The authors should explain the possible reason why they found a lower prevalence of hyperglycaemia compared to other published studies.

Response: Thank you very much for pointing it out. I added an explanation in the text. The median age of the study group was 36 (IQR: 28-45) years old, so young participants could theoretically reduce the prevalence of hyperglycemia in the overall cohort. The prevalence of hyperglycemia was significantly higher in older people, which is not surprising because hyperglycemia and diabetes mellitus are more common in the older population, and the prevalence increases with age. Many other studies have found a similar relationship between increasing age and the prevalence of T2D among people living with HIV.

Comment: Line 190: Consider rephrasing ‘our study cannot be clearly explained’ statement.

Response: It was rephrased as suggested. Thank you for the suggestion.

Comment: Line 192-193: ‘The higher prevalence of hyperlycemia in people with advanced HIV should be interpreted considering previous findings indicating an association between low CD4 count and non-AIDS event’. It sounds there missing phrase? Consider rephrasing it.

Response: The sentence was rephrased: The higher prevalence of hyperglycemia in people with advanced HIV should be interpreted in light of previous findings linking a low CD4 cell count to non-AIDS events. Thank you for pointing it out. More information was added in front to clarify the sentence.

Comment: Under discussion, authors should add the mechanism linking HIV to hyperglycemia.

Response: Thank you for mentioning it. I added the mechanism linking HIV to hyperglycemia in the discussion.

Comment: Other factors such as obesity/ high BMI, and lifestyle that may contribute to hyperglycaemia/impaired fasting glucose were not assessed in this study. Should be highlighted as a limitation.

Response: Sentence: “Other factors such as obesity/ high BMI, and lifestyle that may contribute to hyperglycemia/impaired fasting glucose were not assessed in this study” was added in limitations. Thank you very much for your advice.

General comment:

Authors should leave the space between the last word and reference consistent throughout the manuscript.

Modify the font type for references

Response: It was fixed as suggested. Thank you very much for your advice.

Reviewer #1: Yes: Emma M. Kileel

Reviewer #2: No

Editor

Response: Thank you very much. It was formatted based on links.

Thank you for your thoughtful comment. It was added in method section.

Response: Our research did not have any grants. The author(s) received no specific funding for this work.

“Many thanks to all the doctors and staff of the Infectious Diseases, AIDS & Clinical Immunology Research Center who support the implementation of the HIV/AIDS Program”

“The study has been conducted with internal resources of Infectious Diseases, AIDS & Clinical Immunology Research Center”

Response: The author(s) received no specific funding for this work. was added in the cover letter. Information in acknowledgements was updated. Thank you for your thoughtful comments, suggestions and advice.

“The authors have no conflicts of interest to disclose.”

Please complete your Competing Interests on the online submission form to state any Competing Interests. If you have no competing interests, please state "The authors have declared that no competing interests exist."", as detailed online in our guide for authors at http://journals.plos.org/plosone/s/submit-now

Response: ‘’The authors have declared that no competing interests exist’’ was added in the cover letter. Thank you for your advice.

Response : Thank you very much for the important information. Here are URL and DOI from protocols.io

URL: https://www.protocols.io/private/F851E67044A111EDB19C0A58A9FEAC02

DOI: dx.doi.org/10.17504/protocols.io.3byl4jke8lo5/v1

Response: Full Ethics Statement was added in the methods section. thank you for your help.

Response: References were double-checked. Thank you for your suggestions.

Sincerely 

Tea Borkowska

---

## [Decision Letter · Decision Letter 1]

13 Oct 2022

The prevalence of hyperglycemia and its impact on mortality among people living with HIV in Georgia

PONE-D-22-13596R1

Dear Dr. Borkowska,

We’re pleased to inform you that your manuscript has been judged scientifically suitable for publication and will be formally accepted for publication once it meets all outstanding technical requirements.

Kind regards,

Cristian Apetrei, MD, PhD

Academic Editor

PLOS ONE

Additional Editor Comments (optional):

Reviewers' comments:

Reviewer's Responses to Questions

**Comments to the Author**

1. If the authors have adequately addressed your comments raised in a previous round of review and you feel that this manuscript is now acceptable for publication, you may indicate that here to bypass the “Comments to the Author” section, enter your conflict of interest statement in the “Confidential to Editor” section, and submit your "Accept" recommendation.

Reviewer #1: (No Response)

Reviewer #2: All comments have been addressed

2. Is the manuscript technically sound, and do the data support the conclusions?

Reviewer #1: Yes

Reviewer #2: Yes

3. Has the statistical analysis been performed appropriately and rigorously? 

Reviewer #1: Yes

Reviewer #2: Yes

4. Have the authors made all data underlying the findings in their manuscript fully available?

Reviewer #1: Yes

Reviewer #2: Yes

5. Is the manuscript presented in an intelligible fashion and written in standard English?

Reviewer #1: Yes

Reviewer #2: Yes

6. Review Comments to the Author

Reviewer #1: Thank you to the authors for taking the time to address the previous comments. The authors clarified that glucose measurements used in analyses were measured prior to ART initiation, which was a primary question/concern. Overall, the results of this analysis are clear and have important implications re screening for hyperglycemia at the point of entry into HIV care. The content of this paper is sound, however there are few remaining formatting edits that would help with the readability of the manuscript.

Overall comments:

How hyperglycemia was defined in this study, lines 94 - 103, remains slightly unclear. Consider restructuring by starting with a clear statement that that hyperglycemic status was based off of glucose measures at the time of entry into HIV care (i.e., prior to ART initiation), according to fasting status and the ADA criteria referenced. The authors could then follow with how they defined postprandial glucose measurements, and finally, end with the sentence on lines 95-97, regarding the referment of patients with hyperglycemia to relevant specialists.

The authors abbreviate people living with HIV to PLWH in the introduction section of the manuscript, therefore any future mentions of 'people living with HIV' can be replaced with PLWH.

Minor comments:

Line 59: Change areon to are on

Line 68: Change 'The longer ART exposure is linked...' to 'Longer ART exposure is linked'

Line 87: Change studyincluded to study included

Line 87: Consider changing "The study included adults (age >=18 years) living with HIV diagnosed in 2012-2018 and followed through to 2020" to "The study included adults (age >= 18 years) diagnosed with HIV between 2012-2018 and followed through to 2020."

Line 93: Change allconfirmed to all confirmed

Line 107: Change includingall to including all

Line 109: For consistency, consider changing 'AIDS-related deaths' to just 'AIDS-related' or, changing 'non-AIDS' to 'non-AIDS-related deaths'

Line 136: Authors state "AIDS was documented in 1189 (40.8%) patients", consider changing (if accurate) to "History of AIDS was documented in 1189 (40.8%) patients."

Line 137: Authors state "Every single patient has begun ART." Consider changing to "Every single patient initiated ART".

Line 143: Suggest changing "fasting glucose" to "fasting status" and further suggest changing 'cases' to 'patients'

Line 144: Authors state: "Two hundred and forty-two (8.3%) patients had hyperglycemia, increasing prevalence by age", consider changing to "Overall, 242 (8.3%) patients had hyperglycemia at entry into HIV care, with an increasing prevalence by age"

Reviewer #2: Thank you for taking time to addressing the comments. I am looking forward seeing your paper published.

7. PLOS authors have the option to publish the peer review history of their article (what does this mean?). If published, this will include your full peer review and any attached files.

Reviewer #1: **Yes: **Emma Kileel

Reviewer #2: No

---

## [Editor Report · Acceptance letter]

19 Oct 2022

PONE-D-22-13596R1 

The prevalence of hyperglycemia and its impact on mortality among people living with HIV in Georgia 

Dear Dr. Borkowska:

I'm pleased to inform you that your manuscript has been deemed suitable for publication in PLOS ONE. Congratulations! Your manuscript is now with our production department. 

Kind regards, 

on behalf of

Dr. Cristian Apetrei 

Academic Editor

PLOS ONE